# Assessing the Viability of Segmental Aneuploid Embryos: A Chromosomal Concordance Study of 175 Human Blastocysts

**DOI:** 10.3390/ijms26115284

**Published:** 2025-05-30

**Authors:** En-Hui Cheng, Hui-Hsin Shih, Tsung-Hsien Lee, Pin-Yao Lin, Tzu-Ning Yu, Chun-Chia Huang, Maw-Sheng Lee, Chun-I Lee

**Affiliations:** 1Genetic Diagnosis Laboratory, Lee Women’s Hospital, Taichung 40652, Taiwan; enhuicheng@ivftaiwan.com (E.-H.C.); huihsinshih@ivftaiwan.com (H.-H.S.); 2Post Baccalaureate Medicine, National Chung Hsing University, Taichung 40227, Taiwan; 3Institute of Medicine, Chung Shan Medical University, Taichung 40201, Taiwan; jackth.lee@gmail.com (T.-H.L.); msleephd@gmail.com (M.-S.L.); 4Division of Infertility, Lee Women’s Hospital, Taichung 40402, Taiwan; ningsyu@gmail.com (T.-N.Y.); agarhuang@gmail.com (C.-C.H.); 5Department of Obstetrics and Gynecology, Chung Shan Medical University Hospital, Taichung 40201, Taiwan; 6Department of Obstetrics and Gynecology, School of Medicine, Chung Shan Medical University, Taichung 40201, Taiwan

**Keywords:** preimplantation genetic testing for aneuploidy (PGT-A), segmental mosaicism (Seg-M), segmental aneuploidy (Seg-A), trophectoderm (TE) biopsy

## Abstract

Preimplantation genetic testing for aneuploidy (PGT-A) is widely used to select euploid embryos for in vitro fertilization (IVF), but its accuracy in predicting the implantation potential for full segmental aneuploid (Seg-A) embryos remains unclear. In this study, we investigated chromosomal concordance between clinically biopsied trophectoderm (TE) and inner cell mass (ICM) in 175 donated blastocysts, which comprised those clinically diagnosed as euploid (13), Seg-A (36), segmental mosaicism (Seg-M) (60), whole-chromosome aneuploid (Who-A) (52), and whole-chromosome mosaicism (14). Using next-generation sequencing (NGS), we found that TE–ICM concordance rates were higher for euploid (85%) and Who-A (94%) embryos but significantly lower for Seg-A (25%) and Seg-M embryos (33%). For Seg-A, the euploidy rate in the ICM was 19% and the euploidy rate in the ICM was 63% for Seg-M. These low concordance rates may be due to technical and biological artifacts of PGT-A for Seg-A. Despite the significant discordance between TE and ICM, a subset of Seg-A embryos demonstrated euploidy. While clinically diagnosed euploid embryos remain the preferred choice, Seg-A embryos should be considered as having implantation potential. In particular, Seg-A results should be clearly distinguished from Who-A results and not routinely categorically discarded. Further research is required to refine the selection criteria, aided by parental karyotyping or re-biopsy, and to develop more reliable embryo assessment methods to ensure the accurate evaluation of reproductive potential and support shared decision making between doctors and patients.

## 1. Introduction

Preimplantation genetic testing for aneuploidy (PGT-A) is a technique that is widely used in assisted reproduction to select euploid embryos for transfer, thereby improving implantation rates and reducing miscarriage risks in in vitro fertilization (IVF) [1,2,3]. However, the presence of segmental abnormality, either full or mosaic, poses a significant challenge in embryo selection, as its clinical implications remain unclear [4,5,6]. Segmental aneuploidy (Seg-A) is characterized by partial chromosomal gain or loss. These structural deviations can be either de novo changes caused by a meiotic error in gametogenesis (resulting in Seg-A), mitotic errors caused by early embryonic cleavage (resulting in segmental mosaicism, Seg-M), or inherited from parent carriers of reciprocal translocations [7].

Reported incidences of Seg-A diagnosed in PGT-A blastocysts diverge from approximately 3.1 to 15.6% [8]. When Seg-A is detected in trophectoderm (TE) biopsies, this creates difficulty in determining whether the embryo should be transferred. The use of embryos diagnosed with Seg-A may potentially result in failed implantation, miscarriage, or a chromosomally abnormal baby, but may also result in a healthy live birth [9]. Many PGT laboratories consider these results to be abnormal and advise against transferring the equivalent embryos. However, studies on concordance rates between biopsied TE and inner cell mass (ICM) for Seg-A and Seg-M are limited, and the rates have not been clearly defined [8,10,11,12,13,14].

The current guidelines of the American Society for Reproductive Medicine (ASRM) and the Preimplantation Genetic Diagnosis International Society (PGDIS) provide recommendations for mosaic embryo transfers but lack clear directives for Seg-A embryos due to limited clinical and laboratory data [15,16,17,18]. To address these uncertainties, this study evaluates the chromosomal concordance between biopsied TE and ICM in embryos diagnosed with Seg-A and assesses their implantation potential. By analyzing 175 donated blastocysts (including 36 Seg-A and 60 Seg-M), we aim to provide critical insights into whether Seg-A embryos can support normal development and inform clinical decision making for embryo transfers. To the best of our knowledge, this is the largest cohort of blastocysts used to study Seg-A with respect to TE–ICM concordance reported in the literature. Our findings may contribute to refining embryo selection strategies for Seg-A, particularly in cases when no euploid embryos are available.

## 2. Results

### 2.1. Overall Concordance and Discordance Analysis

In this study, we analyzed 175 blastocysts donated by 89 couples to evaluate the implantation potential of Seg-A using original clinical and ICM biopsies. The original TE biopsy results classified these blastocysts as follows: 13 euploid, 74 mosaic, and 88 aneuploid. Of the 74 mosaic blastocysts, 14 exhibited whole-chromosome mosaicism (Who-M), while 60 displayed Seg-M; of the 88 aneuploid blastocysts, 52 showed whole-chromosome aneuploidy (Who-A) and 36 exhibited Seg-A (Figure 1A). To assess chromosomal concordance, we compared the original TE biopsy results with ICM findings from subsequent analyses (Figure 1B).

Patient characteristics are summarized in Table 1. The percentages of biopsies performed on day 5 and day 6 were 69.1% (121/175) and 30.9 (54/175), respectively. According to the SART (Society for Assisted Reproductive Technology) grading system, the biopsied blastocysts were classified by quality. The percentages of good, fair, and poor blastocysts were 13.1% (23/175), 82.9% (145/175), and 4% (7/175), respectively (Table 1).

### 2.2. Euploidy and Aneuploidy Concordance

The analysis revealed an overall chromosome concordance rate of 55% between the original TE and ICM samples. Specifically, the concordance rates were 85% (11/13) for euploidy, 25% (9/36) for Seg-A, 33% (20/60) for Seg-M, 14% (2/14) for Who-M, and 94% (49/52) for Who-A (Figure 2). These results confirm that euploid and Who-A embryos exhibited high TE–ICM concordance, whereas Seg-A and mosaic embryos showed significantly lower concordance rates, highlighting challenges in their accurate classification.

The ICM euploidy rate was 85% (11/13) for originally euploid TE samples but 0% (0/52) for originally Who-A TE samples. For Seg-A embryos, the ICM euploidy rate was 19% (7/36), while for Seg-M TE samples, it was 63% (38/60) (Figure 2).

These findings emphasize that Seg-A and Seg-M embryos exhibit significantly lower TE–ICM concordance than euploid or Who-A embryos, reinforcing the need for careful embryo selection.

Our results suggest that an embryo classified as Seg-A by PGT-A may still result in a healthy baby, though with a lower likelihood compared to Seg-M.

Diagrams of PGT-A results for Seg-A or Seg-M from the original TE biopsy and ICM showing concordance or discordance are presented in Figure 3. These representative examples illustrate both concordant and discordant findings between TE and ICM in segmental aneuploidy and mosaicism.

### 2.3. Sensitivity and Specificity of Embryo Implantation Potential

The embryo euploidy potential of the TE serves as a predictor of reproductive ability. Implantation potential is classified based on whether the embryo exhibits mosaicism below or above the 50% threshold. Embryos with mosaicism <50% were considered to have euploidy potential. The predictive accuracy of TE biopsies in assessing ICM euploidy was evaluated using sensitivity, specificity, positive predictive value (PPV), and negative predictive value (NPV), as shown in Table 2. The sensitivity and specificity of TE biopsy for predicting ICM euploidy were calculated across different chromosomal abnormalities.

For Who-A, the sensitivity was 98.1%, and the specificity was 100%, with a PPV of 100% and an NPV of 92.3% (Table 2). For Seg-A, the sensitivity was 94.4%, while the specificity was lower at 38.7%, with a PPV of 47.2% and an NPV of 92.3% (Table 2). Notably, embryos classified as Seg-A in TE biopsies may still be suitable for transfer, as the ICM can sometimes be euploid despite segmental aneuploidy in the TE. PGT-A results for Seg-A embryos have limited specificity, indicating that some embryos classified as segmental aneuploid in TE biopsies may, in fact, be euploid in the ICM. This supports the notion that certain Seg-A embryos may still have reproductive potential and should not be categorically excluded from transfer considerations.

The table presents the sensitivity, specificity, PPV, and NPV of various PGT-A results in predicting the euploidy status of the ICM. The classification of implantation potential is based on whether the embryo exhibits mosaicism below or above 50%. Sensitivity = TP/(TP + FN): the ability of the test to correctly identify euploid ICM. Specificity = TN/(TN + FP): the ability of the test to correctly identify non-euploid ICM. Positive predictive value (PPV) = TP/(TP + FP): the probability that a PGT-A result indicating aneuploidy or mosaicism accurately predicts the ICM status. Negative predictive value (NPV) = TN/(TN + FN): the probability that a PGT-A result indicating euploidy is actually correct. Abbreviations: TP = true positive, FP = false positive, FN = false negative, TN = true negative.

### 2.4. Correlation Between ICM Status and TE Biopsy in Seg-A and Seg-M

The NGS results for Seg-A in the TE biopsy and ICM as identified by PGT-A are presented in Table 3. This dataset includes 36 embryos diagnosed with Seg-A, detailing their embryo grade, segment size (Mb), chromosomal regions affected (gain or loss), and corresponding chromosomal status in the ICM. The average Seg-A fragment size detected in the TE biopsy was 40.2 Mb, ranging from 10 Mb to 140 Mb. Notably, 97% (35/36) of Seg-A cases involved only a single chromosomal segment. Embryos classified as euploid in the ICM had an average original Seg-A segment size of 38.6 Mb, all of which were single-segment abnormalities. Due to the limited sample size, no significant correlation was observed between the occurrence of Seg-A in different chromosomes, segment size, and ICM chromosomal status.

To further explore the relationship between TE biopsy and ICM chromosomal status, an additional dataset of 60 embryos diagnosed with Seg-M in the TE biopsy is shown in Appendix A. Among these cases, embryos classified as euploid in the ICM exhibited mosaic losses or gains affecting single or multiple small chromosomal segments, but no clear correlation was identified between specific chromosomal regions and ICM status. Interestingly, mosaic losses were more frequently observed than mosaic gains, with the most commonly affected chromosomal regions including 5q, 7q, 11q, and Xq. Overall, the results suggest a high degree of discordance between TE biopsies and the ICM chromosomal status in embryos diagnosed with both Seg-A and Seg-M. Despite the presence of segmental aneuploidy or mosaicism in the TE biopsy, a certain portion of embryos exhibited euploidy in the ICM.

In Seg-A embryos, the majority of abnormalities involved a single chromosomal segment, with no clear correlation between the affected chromosome and ICM status. Similarly, in Seg-M embryos, mosaic losses occurred more frequently than gains, but no distinct pattern of chromosomal involvement was observed (Appendix A). These findings highlight the limitations of PGT-A in accurately predicting the chromosomal integrity of the ICM, particularly for embryos tested with Seg-A and Seg-M. The high TE-ICM discordance underscores the need for cautious interpretation of PGT-A results, especially when segmental abnormalities are detected. Further investigations with larger sample sizes and functional studies are necessary to determine the clinical significance of Seg-A and Seg-M and to optimize embryo selection strategies in assisted reproductive technology.

## 3. Discussion

In this study, we analyzed 175 donated blastocysts after clinical TE biopsy, including 36 diagnosed with Seg-A and 60 with Seg-M. This is the largest number of blastocysts used in the literature so far for studying Seg-A with regard to the concordance of TE biopsy and ICM. We found that the overall TE-ICM concordance rate was 55%, with significantly higher concordance observed for euploid (85%) and Who-A (94%) embryos compared to those diagnosed with Seg-A (25%) or Seg-M (33%). This discrepancy is clinically significant, as it suggests that TE biopsy alone may not accurately represent the chromosomal composition of the ICM in Seg-A embryos compared to euploid and Who-A embryos. This highlights the need for the cautious interpretation of PGT-A results in embryos with segmental abnormalities.

Victor et al. [11] reported high TE-ICM concordance for Who-A embryos (97%) but a low concordance for Seg-A embryos (43%); however, only seven Seg-A blastocysts were studied, and Seg-A and Seg-M were not clearly defined in the study. Girardi et al. [19] found that the positive predictive value of biopsied TE for ICM confirmation was significantly lower for Seg-A compared to Who-A (71% versus 97%, respectively). Of the 53 blastocysts classified as Seg-A, 17 were found to be Seg-A and 36 were Seg-M. Our findings using a larger series reinforce that PGT-A results for Seg-A embryos are less predictive of ICM chromosomal status, making embryo selection more challenging.

As shown in Table 3, a number of embryos originally classified as Seg-A in the TE biopsy were reclassified in the ICM biopsy, reflecting significant TE–ICM discordance; as demonstrated in Figure 2, the concordance rate was 25%. As shown in Appendix A, those embryos originally classified as Seg-M were also reclassified in the ICM biopsy, reflecting further significant TE–ICM discordance. The concordance rate was 33%. The TE-ICM concordance rates for both Seg-A and Seg-M were low. These high discordance rates suggest limitations in the predictive value of a single TE biopsy. Nonetheless, the euploidy rate among Seg-A embryos was 19%, compared to 63% in Seg-M embryos, suggesting that a substantial proportion of these embryos may still have developmental potential. We believe that Seg-A and Seg-M embryos should not be categorically excluded from transfer, especially in the absence of euploid embryos. Instead, we recommend enhanced genetic counseling that includes parental karyotyping to support informed decision making. Re-biopsy should be performed in cases involving critical chromosomal regions.

For Seg-A, the TE-ICM concordance rate was 25%, and the euploidy rate in the ICM was 19%. This suggests that some embryos diagnosed as Seg-A in TE biopsies may have euploid ICMs, further supporting the hypothesis that certain Seg-A embryos may be viable. Some Seg-A embryos may originate from inherited structural rearrangements rather than de novo meiotic errors, and as such, parental karyotyping may be indicated in the selected cases. For carriers of parental structural rearrangement, unbalanced embryos presenting as Seg-A would not be suitable for transfer [20,21]. From a clinical standpoint, our findings may support the re-biopsy of blastocysts in patients who have only produced embryos classified as de novo Seg-A. The use of a second TE biopsy could effectively improve the predictivity of ICM composition and the decision-making procedure. If Seg-A is detected in both biopsies, this is thought to be consistent with a pattern of meiotic origin, and the embryo would not be suitable for transfer.

Although PGT-A guidelines generally discourage transferring embryos with Seg-A, our findings challenge the assumption that all Seg-A embryos lack viability, reinforcing the need for more refined selection criteria. Recently, several live births have been reported following the transfer of embryos previously classified as Seg-A [9,22]. This underscores the potential for some PGT Seg-A embryos to develop into healthy offspring. Further research is needed to determine whether the parental chromosomal background should influence the decision to transfer Seg-A embryos and whether re-biopsy of de novo Seg-A embryos is warranted, as our data suggest that Seg-A embryos should not be categorically discarded routinely.

Seg-M abnormalities are typically of mitotic origin and arise during early embryonic cleavage. Emerging evidence suggests that embryos have the capacity to self-correct chromosomal errors through mechanisms such as the preferential elimination of aneuploid cells or compensatory events within the ICM. For instance, Daughtry et al. reported that embryos respond to chromosomal abnormalities through micronuclei encapsulation, cellular fragmentation, and selection against aneuploid blastomeres [23]. Similarly, Singla et al. demonstrated that aneuploid cells can be selectively eliminated from the embryonic lineage in a p53-dependent manner [24]. Our findings support these biological mechanisms, especially in cases where the detection of Seg-M in TE is not reflected by the ICM chromosomal status.

Given the variable outcomes associated with Seg-M, we recommend that these embryos are not routinely excluded from transfers. Instead, individualized genetic counseling should be offered to help patients understand the associated uncertainties and potential outcomes. While re-biopsy may be considered in selected cases—particularly when the segmental mosaicism involves critical chromosomal regions—it should be approached with caution due to the potential harm caused by additional manipulation. Recent updates from the ASRM and PGDIS suggest that Seg-M embryos may still result in healthy pregnancies [6,15,16,18], although they may have lower implantation success rates. Several studies have also reported successful live births following the transfer of Seg-M embryos [25,26].

Several limitations should also be acknowledged for the detection of Seg-A using PGT. With regard to the technical and biological artifacts of PGT-A, while next-generation sequencing (NGS) can achieve high-resolution chromosomal analysis, it may not capture all mosaic patterns, warranting further validation with single-cell sequencing or advanced embryo diagnostics. Errors may potentially arise from S-phase artifacts that mimic the appearance of Seg-A [27]. Future studies should also investigate the impact of segment size on implantation success, the role of mitotic vs. meiotic origin in Seg-A embryo viability, and clinical outcomes from larger Seg-A embryo datasets.

This study provides valuable insights into the complexity of Seg-A embryos, demonstrating that a subset may develop into healthy live births. While euploid embryos remain the preferred choice, our findings support the consideration of Seg-A embryos in certain clinical scenarios. Refining selection criteria with the help of parental karyotyping or re-biopsy and developing more reliable embryo assessment methods will be critical for optimizing outcomes in assisted reproductive technologies. Given the variability in pregnancy outcomes, clinicians should exercise caution when considering Seg-A or Seg-M embryo transfers, albeit at decreased implantation rates, particularly in cases where euploid embryos are unavailable. Seg-A results should be clearly distinguished from Who-A results to enable the accurate assessment of reproductive potential and support informed, shared decision making between clinicians and patients.

Although our data and other reports suggest that a subset of Seg-A embryos may have reproductive potential [9,19,28], it is important to emphasize that euploid embryos should always remain the first-line choice for transfer due to their significantly higher predictability and implantation outcomes. The transfer of Seg-A embryos should only be considered as a secondary option, particularly in cases where no euploid embryos are available for implantation. This approach aligns with current PGT-A recommendations, which advocate for the cautious use of segmental abnormal embryos given their uncertain developmental competence and lower concordance with ICM results.

Our findings support the notion that, in selected cases—such as when parental karyotyping suggests a structural rearrangement or after confirmatory re-biopsy—some Seg-A embryos may be viable. Nonetheless, the decision to transfer a Seg-A embryo should be made with full informed consent and in the context of limited alternatives, ensuring that patients are aware of the associated risks and reduced predictability.

## 4. Materials and Methods

### 4.1. Ethics Approval and Consent to Participate

This study was approved by the Institutional Review Board of Chung Shan Medical University Hospital (IRB: CS16105). All procedures were conducted in accordance with relevant ethical guidelines and regulatory standards.

### 4.2. Clinical PGT-A and Donated Embryos Using Next Generation Sequence (NGS)

All in vitro fertilization (IVF) cycles were performed at Lee’s Women’s Hospital between April 2017 and March 2020. PGT-A was conducted in the certified genetic diagnostic laboratory. After clinical TE biopsy, embryos were vitrified until the NGS results were obtained. Frozen embryo transfers were performed to achieve pregnancy. After the IVF cycles or live birth deliveries were complete, a total of 175 excess or abnormal blastocysts from 89 couples were donated for this study. Written informed consent was obtained from all participating couples.

### 4.3. Ovarian Stimulation, IVF, and Embryo Culture

Ovarian stimulation was performed as described by Lee et al. [29]. The retrieved oocytes were cultured in Quinn’s Advantage Fertilization Medium (Sage BioPharma, Inc., Trumbull, CT, USA) supplemented with 15% Serum Protein Substitute (SPS; Sage BioPharma) in a triple-gas incubator maintained at 5% CO_2_, 5% O_2_, and 90% N_2_. Following fertilization via conventional IVF or intracytoplasmic sperm injection (ICSI), embryos were cultured in micro-droplets of culture medium supplemented with 15% SPS until day 3. Subsequently, embryos were transferred to a blastocyst medium supplemented with 15% SPS and maintained in a group culture system.

### 4.4. TE Biopsy and Embryo Cryopreservation

On day 4 post-fertilization, a small opening was created in the zona pellucida using a laser pulse to facilitate TE herniation. Expanded blastocysts were biopsied on day 5 or 6. TE biopsy was performed when the blastocyst achieved a grade of 4BC/4CB or higher, as per Gardner’s scoring system [30]. The blastocyst was stabilized using a holding pipette (Humagen, Charlottesville, VA, USA), and 5–10 TE cells were aspirated with a biopsy pipette, utilizing laser pulses of 1.3–1.5 microseconds (OCTAX NaviLase® Vitrolife GmbH, Landshut, Germany) to facilitate cell detachment. Biopsied cells were washed three times in phosphate-buffered saline under a microscope and immediately transferred into RNase/DNase-free PCR tubes for genetic analysis. Following biopsy, blastocysts were cultured in a blastocyst medium supplemented with 15% SPS in a tri-gas incubator until vitrification. Vitrification was performed using the Cryotech vitrification method (Repro-Support Medical Research Centre, Tokyo, Japan) according to the protocol described by Gutnisky et al. [31].

### 4.5. Thawed Blastocysts and Biopsy of ICM

During thawing, blastocysts were placed in pre-warmed 1 M sucrose for 1 min at 37 °C, then transferred to 0.5 M sucrose for 3 min, followed by 10 min in a basic solution (HEPES-buffered tissue culture medium with 20% SPS) at room temperature, changing the solution after 5 min [32]. After thawing, blastocysts were washed in Global medium supplemented with 10% SPS and cultured overnight in the same medium before biopsy. When the blastocyst cavity re-expanded and the ICM could be identified, a re-biopsy was performed on the thawed blastocysts. The blastocyst was placed on a holding pipette with the ICM visible opposite the biopsy pipette (Figure 1B). The ICM was dissected using laser pulses of 1.3–1.5 microseconds, aspirated, and then expelled to prevent mixing with TE cells (Figure 1C). ICM cells were immediately placed in RNase/DNase-free PCR tubes [33].

### 4.6. NGS Protocol for PGT-A

Biopsied samples were placed in a lysis buffer, and genomic DNA was fragmented and amplified (SurePlex DNA Amplification System, Illumina, San Diego, CA, USA), following the manufacturer’s guidelines. Whole-genome amplification products of each sample were prepared according to the VeriSeq PGS workflow (Illumina). The purified DNA library was normalized to quantify each sample’s final pooled amount using library normalization additives and purification beads. Normalized samples were pooled, denatured, and sequenced using the Miseq system and reagents (Miseq v.3, Illumina). Bioinformatics data were analyzed using BlueFuse Multi Software v4.4 (Illumina), and two technicians re-verified the mosaic chromosomes of each sample. Segmental gains or losses were defined as 10 Mb changes in diploid–aneuploid mosaic ratios detected by the high-resolution (h) NGS platform [34]. Raw NGS data were uploaded to the NCBI database.

### 4.7. Data Availability

The NGS raw data that support the findings of this study have been deposited in the NCBI BioProject database with the primary accession code PRJNA989569.

### 4.8. Definition of Chromosome Concordance and Discordance

Using the percentages of diploid–aneuploid mosaicism detected utilizing the hr-NGS platform for clinically biopsied TE cells, blastocysts were categorized into five groups as follows: (1) euploid (Eu) blastocysts with mosaicism levels < 25%; (2) full segmental aneuploid (Seg-A) blastocysts with mosaicism levels < 25%; (3) segmental mosaicism (Seg-M) blastocysts with mosaicism levels between ≥25% and <85%; (4) whole-chromosome aneuploid (Who-A) blastocysts with mosaicism levels > 85%; and (5) whole-chromosome mosaicism (Who-M) blastocysts with mosaicism levels between ≥25% and <85%. Concordance was defined as the equivalence between the initial clinical biopsy sample and those of cells from the ICM (Figure 1C). If the PGT-A results from the original TE sample and those from the ICM indicated different chromosome statuses, the chromosomes were considered discordant.

### 4.9. Evaluation Sensitivity and Specificity of Embryo Implantation Potential

Positive and negative predictive value were utilized to assess embryo potential based on the euploidy status of the ICM. Viotti et al. [35] analyzed clinical outcomes from 1000 mosaic embryo transfers to establish an embryo ranking system for clinical application. A threshold of 50% mosaicism was proposed as a determinant of implantation success, with embryos classified into "low and high implantation potential" groups based on mosaicism levels. Embryos exhibiting mosaicism below 50% were considered to have higher euploid potential due to the lower proportion of abnormal cells, which correlated with improved implantation and ongoing pregnancy rates. The 50% threshold is clinically significant, as data indicate superior outcomes for embryos with lower mosaicism levels. Whole-chromosome mosaic embryos with <50% aneuploid cells demonstrated significantly better implantation and pregnancy outcomes than those with >50% aneuploid cells. Consequently, embryos with <50% mosaicism were considered to have high implantation potential, whereas those with mosaicism levels exceeding 50% were classified as having low implantation potential.

### 4.10. Statistical Analysis

We calculated the sensitivity, specificity, positive predictive value (PPV), and negative predictive value (NPV) of different PGT-A results from the original TE biopsy to predict ICM euploidy. Statistical comparisons of categorical outcomes (e.g., chromosomal status between TE and ICM) were performed using the chi-square test in SPSS version 22.0. Pairwise comparisons were evaluated using chi-square post hoc analysis with Bonferroni correction for multiple testing where appropriate. A *p* value < 0.05 was considered statistically significant.

## 5. Conclusions

In conclusion, this study provides valuable insights into the ambiguity of PGT-A data regarding segmental aneuploidy, emphasizing the need for cautious interpretation of the results and the development of more reliable diagnostic tools, particularly with regard to the discordance between trophectoderm and ICM results in Seg-A and Seg-M embryos. While euploid embryos remain the preferred and most reliable choice for embryo transfer, our findings support the careful consideration of Seg-A embryos in selected clinical scenarios, especially when no euploid embryos are available. Key strategies, including parental karyotyping, confirmatory re-biopsy, and the development of more accurate embryo assessment tools, will be essential in refining embryo selection and improving reproductive outcomes. These results reinforce the need for individualized counseling and thoughtful decision making in assisted reproductive technologies, rather than the routine exclusion of embryos with segmental abnormalities.

## Figures and Tables

**Figure 1 ijms-26-05284-f001:**
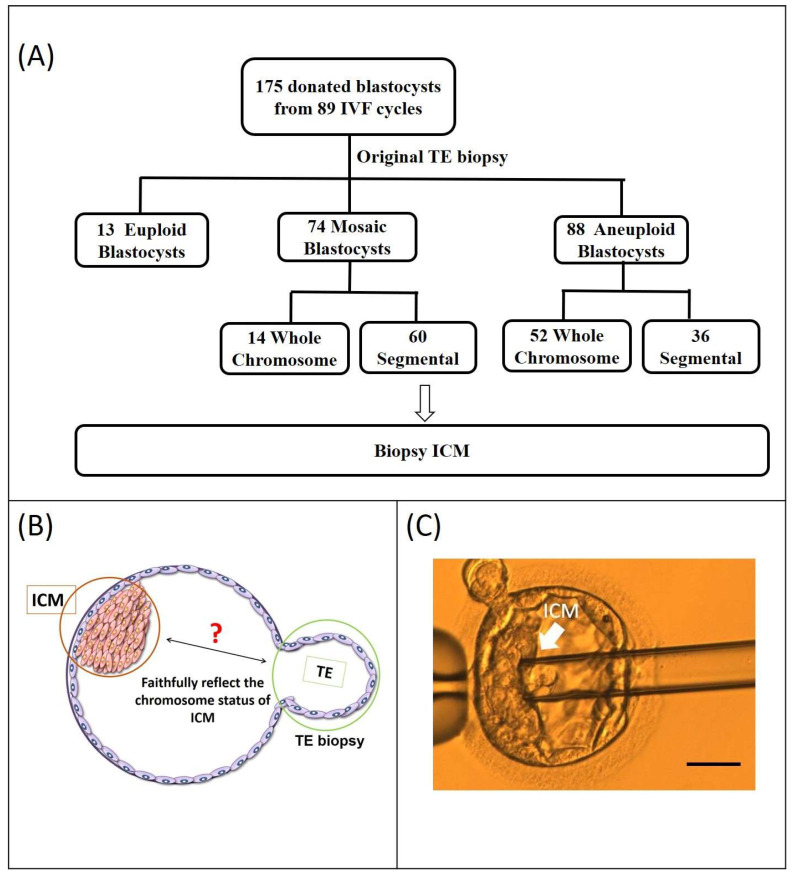
Study flowchart. (**A**) A total of 175 blastocysts were donated by 89 couples. The original PGT-A results of the donated blastocysts are displayed in the second row. PGT-A results from the original trophectoderm (TE) biopsy indicated 13 euploid, 74 mosaic, and 88 aneuploid blastocysts. Of the 74 mosaic blastocysts, 14 exhibited whole-chromosome mosaicism, while 60 displayed segmental mosaicism; of the 88 aneuploid blastocysts, 52 showed whole-chromosome aneuploidy, and 36 exhibited Seg-A. Chromosome status of mosaic and aneuploidy blastocysts is displayed in the third row. (**B**) The location of the ICM re-biopsy and initial clinical TE biopsy sites (ICM: inner cell mass; TE: trophectoderm). (**C**) Isolation of the ICM from a blastocyst. The blastocyst is positioned for micromanipulation. The ICM, marked by the arrow, is targeted for isolation and then aspirated into the micropipette, demonstrating the successful extraction process. (scale bar = 50 μm)

**Figure 2 ijms-26-05284-f002:**
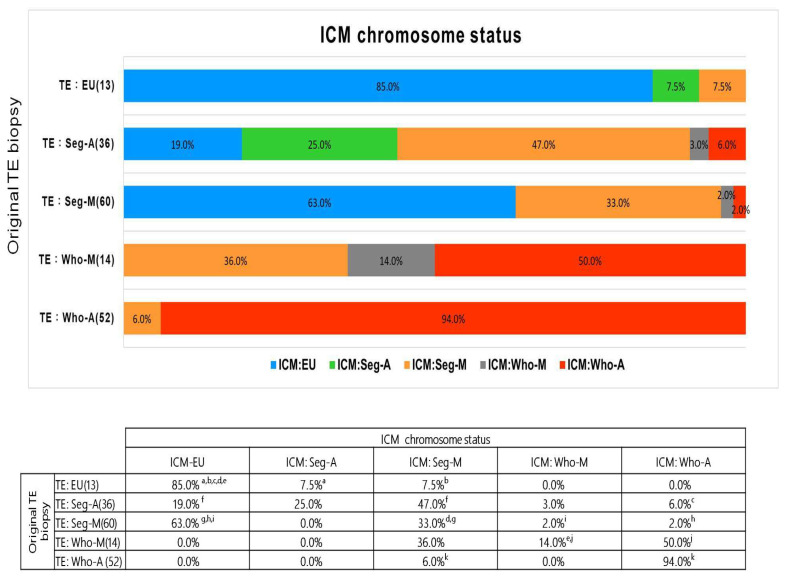
Concordance between original TE biopsy and ICM chromosome status. Bar plots show the distribution of ICM chromosome status among embryos grouped by original TE biopsy results. Superscript letters (a–k) indicate statistically significant pairwise differences between groups, based on the chi-square test. Each letter represents a specific comparison; identical letters denote two values that are significantly different from each other (*p* < 0.05).

**Figure 3 ijms-26-05284-f003:**
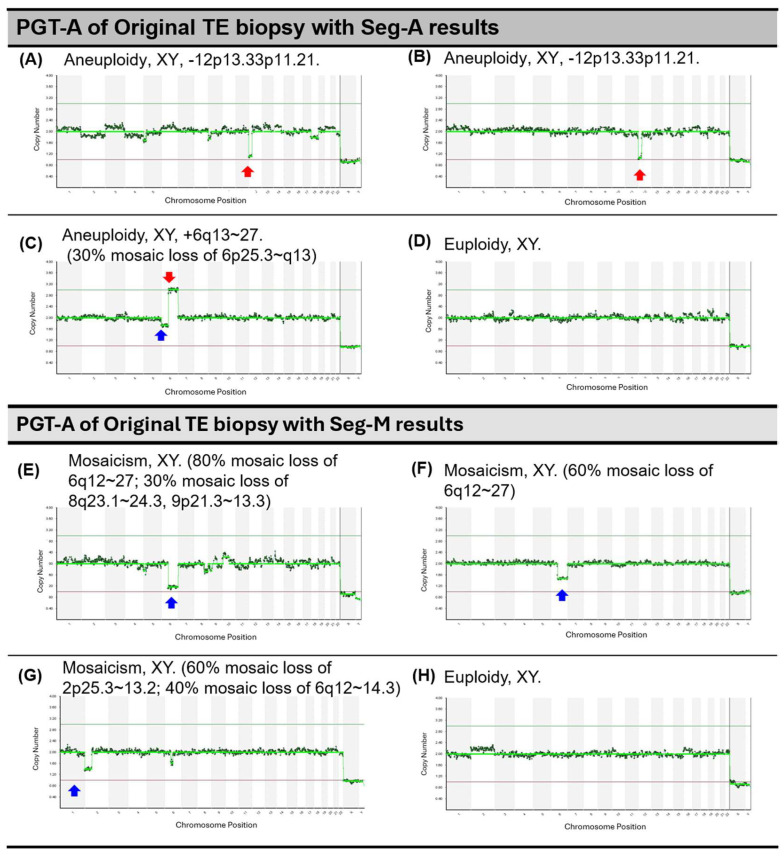
Concordance and discordance of PGT-A results between original TE biopsy and ICM. This figure presents the concordance and discordance of segmental aneuploidy (Seg-A) and segmental mosaicism (Seg-M) between the original trophectoderm (TE) biopsy and the inner cell mass (ICM). (**A**,**B**) Concordance of Seg-A: both the original TE biopsy (**A**) and ICM (**B**) exhibit the same segmental aneuploidy (loss of 12p13.33p11.21), as indicated by the red arrows. (**C**,**D**) Discordance of Seg-A: the original TE biopsy (**C**) shows segmental aneuploidy (gain of 6q13–27 and 30% mosaic loss of 6p25.3–q13, marked by red and blue arrows), while the ICM (**D**) is euploid. (**E**,**F**) Concordance of Seg-M: the original TE biopsy (**E**) and ICM (**F**) both show mosaicism, with consistent segmental mosaic loss (6q12–27) and additional mosaic loss regions in (**E**), indicated by blue arrows. (**G**,**H**) Discordance of Seg-M: the original TE biopsy (**G**) shows 60% mosaic loss of 2p25.3–13.2 and 40% mosaic loss of 6q12–14.3, while the ICM (**H**) is euploid.

**Table 1 ijms-26-05284-t001:** Characteristics of patients who donated blastocysts.

Patient Characteristics ^a^	
IVF cycles	89
Mean age ± SD (yr)	36.0 ± 4.1 ^b^
BMI (kg/m^2^)	21.7 ± 3.1 ^b^
AMH (ng/mL)	4.9 ± 3.3 ^b^
PGT-A Indicator	
Egg donation	1
Advanced maternal age (≧38 years)	24
Repeated miscarriage (≧2 times)	16
Repeat implantation failure (≧3 times)	29
Male factor (severe oligospermia)	1
Others ^c^	18
Total Blastocysts	175
Biopsied on day 5	69.1% (121/175)
Biopsied on day 6	30.9% (54/175)
Embryo Quality (SART grading system)	
Good	13.1% (23/175)
Fair	82.9% (145/175)
Poor	4% (7/175)

a: Abbreviations: IVF = in vitro fertilization; BMI = body mass index; AMH = anti-Müllerian hormone; PGT-A = preimplantation genetic testing for aneuploidy. b: Mean ± standard deviation (SD). c: Other factors included unexplained infertility, tubal factors, preimplantation genetic testing for monogenic disorders, and aneuploidy.

**Table 2 ijms-26-05284-t002:** Sensitivity, specificity, positive predictive value (PPV), and negative predictive value (NPV) of different PGT-A results for predicting euploidy ICM in the original trophectoderm (TE) biopsy.

Original TE	No of Implantation Potential		Sensitivity	Specificity	PPV	NPV
	Low (TP)	High (FP)				
Euploidy (reference)	1 (FN)	12 (TN)	-	-	-	
Seg-A	17	19	94.4%	38.7%	47.2%	92.3%
Seg-M	5	55	83.3%	17.9%	8.3%	92.3%
Who-M	7	7	87.5%	63.2%	50.0%	92.3%
Who-A	52	0	98.1%	100.0%	100.0%	92.3%

**Table 3 ijms-26-05284-t003:** NGS results for segmental aneuploidy of original biopsy and inner cell mass using PGT-A.

No.	Embryo Quality	Seg-A of Original TE Biopsy	Estimated Size of Seg-A (Mb)	PGT-A Classification of ICM	NGS Results for ICM
1	4BB	gain 6q13~27.	80	EU	Euploidy
2	5BB	loss 6q25.3~27.	50	EU	Euploidy
3	5BB	gain 6q15~27.	60	EU	Euploidy
4	4AA	loss 7p22.3~15.1.	30	EU	Euploidy
5	3AB	gain 11p15.5~15.2.	15	EU	Euploidy
6	4AB	loss 13q31.1~34.	25	EU	Euploidy
7	5BB	gain 16p13.3~12.3.	10	EU	Euploidy
8	5BB	gain 1q12~44.	130	Seg-M	Mosaicism, XY; 30% mosaic gain of 11p11.2~q12.2; 30% mosaic loss of 1q21.2~44
9	4BB	gain 1q32.1~44.	95	Seg-M	Mosaicism, XX; 30% mosaic gain of 8q24.12~24.3; 30% mosaic loss of 2q32.1~37.3 and 9p24.3~21.1
10	5BB	gain 4p14~q13.1.	140	Seg-M	Mosaicism, XY; 40% mosaic loss of 5p15.33~15.2; 30% mosaic loss of 2p25.3~23.3
11	5CB	loss 4q31.3~35.2.	50	Seg-M	Mosaicism, XY; 60% mosaic loss of 4q31.3~4q35.2; 30% mosaic loss of 10q26.13~10q26.3
12	5BC	loss 5q12.1~14.1.	15	Seg-M	Mosaicism, XX; 80% mosaic loss of 5q12.1~14.1
13	5BC	loss 6p25.3~21.33.	20	Seg-M	Mosaicism, XX; 50% mosaic loss of 6p25.3~6p21.33, 6q26~6q27; 30% mosaic loss of 2p25.3~2p24.1
14	5BB	loss 6q25.3~27.	50	Seg-M	Mosaicism, XY; 30% mosaic loss of chr6
15	5BB	gain 7q21.12~36.3.	35	Seg-M	Mosaicism, XX; 30% mosaic gain of 11q24.2~25
16	4BB	loss 7q33~36.3	23	Seg-M	Mosaicism, XX; 80% mosaic loss of 7q33~36.3
17	5BC	loss 7q31.31~36.3.	10	Seg-M	Mosaicism, XY; 60% mosaic gain of 7q31.1~31.31; 60% mosaic loss of 7q31.31~36.3
18	5BA	gain 8q22.1~24.3.	40	Seg-M	Mosaicism, XY; 40% mosaic loss of 5p15.33~15.2, 6q26~27; 30% mosaic gain of 19p13.3~12; 30% mosaic loss of 14q32.12~32.33
19	5BB	gain 9p24.3~23.	20	Seg-M	Mosaicism, XY; 30% mosaic loss of 9p24.3~23
20	4BB	gain 12q13.11~24.33.	60	Seg-M	Mosaicism, XX; 30% mosaic loss of 5p15.33~15.2, 7q35~36.3, 13q33.1~34
21	3BB	gain 13q12.11~21.32.	30	Seg-M	Mosaicism, XY; 40% mosaic loss of 13q21.32~34; 30% mosaic gain of 19p13.3~12
22	3BB	gain 16p13.3~q11.2.	63	Seg-M	Mosaicism, XX; 80% mosaic gain of 16p13.3~q11.2
23	5BA	gain 18p11.32~q11.2	45	Seg-M	Mosaicism, XY; 30% mosaic gain of 19q13.11~13.42, 19p13.3~12; 30% mosaic loss of 2p25.3~24.1
24	4BA	gain 22q12.2~13.33.	10	Seg-M	Mosaicism, XY; 30% mosaic loss of 5p15.33~15.1
25	4BB	loss Xp22.33~11.3.	25	Seg-M	Mosaicism, XX; 60% mosaic loss of Xp22.33~Xp11.3; 30% mosaic gain of 4p16.1~14 and 19p13.3~13.11)
26	4BB	loss 3p26.3~14.1.	35	Seg-A	Aneuploidy, XY; loss 3p26.3~14.1.
27	3BB	loss 4q22.3~35.2.	75	Seg-A	Aneuploidy, XX; loss 4q22.2~35.2.
28	5BB	loss 5q34~35.3.	10	Seg-A	Aneuploidy, XY; loss 5q34~35.3.
29	5BB	loss 8q21.11~24.3.	30	Seg-A	Aneuploidy, XX; gain 3q27.3~3q29, loss 8q21.11~24.3.
30	5BC	loss 8q24.12~24.3.	10	Seg-A	Aneuploidy, XX; loss 8q24.12~24.3.
31	6BB	loss 12p13.33~11.21.	35	Seg-A	Aneuploidy, XY; loss 12p13.33~11.21.
32	5BB	loss 17q22~24.3.	20	Seg-A	Aneuploidy, XY; loss 17q22~24.3.
33	5BC	loss 18q22.1~23.	10	Seg-A	Aneuploidy, XY; loss 18q22.1~23.
34	5BC	loss Xq25~28.	20	Seg-A	Aneuploidy, XX; loss Xq25~28.
35	5BB	loss 4q25~35.2.	60	Who-A	Aneuploidy, XY; loss chr 4.
36	5BA	gain 14q23.2~24.3, loss 21q22.11~22.3.	10 10	Who-A	Aneuploidy, XY; loss chr21.

## Data Availability

The data for this study are available and can be accessed via the National Center for Biotechnology Information (NCBI) BioProject database, which can be found at https://www.ncbi.nlm.nih.gov/bioproject/?term=PRJNA989569, under the accession number PRJNA989569 (accessed on 30 June 2023).

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
