# Peer review of "Assessing the Viability of Segmental Aneuploid Embryos: A Chromosomal Concordance Study of 175 Human Blastocysts"

_ijms, 2025, doi:10.3390/ijms26115284_

Round 1

Reviewer 1 Report

Comments and Suggestions for Authors

The article by Cheng et al. describes the analysis of 175 human blastocysts discarded after IVF to find out the concordance between trophoectoderm (TE) and inner cell mass (ICM) preimplantation genetic testing for aneuplody (by NGS) in order to select euploid embryos for IVF. Critical findings are (1) there are high concordance rates between TE and ICM for embryos classified as euploid or as whole-cromosome aneuploid by TE testing; (2) correlations are quite poor for embryos classified by TE testing either as fully segmental aneuploid or for any mosaic embryos; (3) TE testing by NGS is very good at correctly predicting euploidy (NPV of 92,3%); (4) embryos classified as fully segmental aneuoploif o segmental mosaic still have significantly high levels of euploidy in the ICM (19% and 63%), suggesting a possibly successful  IVF if used for implantation. The authors suggest re-examination of embryos (either by re-biopsy or parental karyotyping) with the latter classification in cases of couples who were unable to generate euploid embryos.

The article will impact guidelines for decision making in the IVF setting. It should be revised by a native English speaker to correct minor language inconsistencies.

I have only two concern:

(1) Table 3: Re-examination of SegM and SegA embryos (table 3) shows little concordance between TE and ICM biopsies (eg. embryos 1-11, 15, 18-21, 23-24...) for embryos reclassfied as SegM. Why? Should we consider not trusting any results for SegM embryos and ask for re-examination? This should be discussed.

(2) Conclusions: I believe that the advice provided in lines 266-269 of the discussion ("While euploid embryos remain the preferred choice, our findings support the consideration of Seg-A embryos in certain clinical scenarios. Refining selection criteria by the help of parental karyotyping or re-biopsy and developing more reliable embryo assessment methods will be critical in optimizing outcomes in assisted reproductive technologies") is one of the most important take-home messages of this paper. In my opinion, this should be stated forcefully in the conclusions.

(2)

Comments on the Quality of English Language

Please revise carefully the text. There are sentences without verb, wrong uses of "euploidy" for "euploid", etc.

Author Response

Dear reviewer 1,

We sincerely appreciate your valuable comments. Please find below our responses and the corresponding revisions made to the manuscript. In response, we have carefully revised the manuscript and addressed all comments in a detailed point-by-point reply.
Comments of Review 1

The article by Cheng et al. describes the analysis of 175 human blastocysts discarded after IVF to find out the concordance between trophectoderm (TE) and inner cell mass (ICM) preimplantation genetic testing for aneuploidy (by NGS) in order to select euploid embryos for IVF. Critical findings are (1) there are high concordance rates between TE and ICM for embryos classified as euploid or as whole-chromosome aneuploid by TE testing; (2) correlations are quite poor for embryos classified by TE testing either as fully segmental aneuploid or for any mosaic embryos; (3) TE testing by NGS is very good at correctly predicting euploidy (NPV of 92,3%); (4) embryos classified as fully segmental aneuploid or segmental mosaic still have significantly high levels of euploidy in the ICM (19% and 63%), suggesting a possibly successful  IVF if used for implantation. The authors suggest re-examination of embryos (either by re-biopsy or parental karyotyping) with the latter classification in cases of couples who were unable to generate euploid embryos.

The article will impact guidelines for decision making in the IVF setting. It should be revised by a native English speaker to correct minor language inconsistencies.

Response:

Thank you for your helpful comments regarding the language and clarity. We have carefully reviewed and revised the manuscript for grammar and typographical errors and sentence clarity. The manuscript has now been edited by a native English speaker with academic writing experience. We believe that the revised version has a significantly improved clarity and language quality. We sincerely appreciate your feedback, which helped improve the readability of our manuscript.

I have only two concern:

(1) Table 3: Re-examination of SegM and SegA embryos (table 3) shows little concordance between TE and ICM biopsies (eg. embryos 1-11, 15, 18-21, 23-24...) for embryos reclassfied as SegM. Why? Should we consider not trusting any results for SegM embryos and ask for re-examination? This should be discussed.

Response:

Thank you for this important observation. Yes, as shown in Table 3, a number of embryos originally classified as Seg-A in the TE biopsy were reclassified in the ICM, reflecting significant TE–ICM discordance. As demonstrated in Figure 2, the concordance rate was 25%. Those originally classified as Seg-M were reclassified in the ICM, shown in Supplementary Table 1, also reflecting significant TE–ICM discordance. The concordance rate was 33%. The TE–ICM concordance rates for both Seg-A and Seg-M were low. These high discordances suggest limitations in the predictive value of a single TE biopsy. Nonetheless, the euploidy rate was 19% and 63% among Seg-A and Seg-M embryos, respectively, suggesting that a substantial proportion of these embryos may still have developmental potential. We believe that Seg-A and Seg-M embryos should not be categorically excluded from transfer, especially in the absence of euploid embryos. Instead, we recommend enhanced genetic counseling, including parental karyotyping, to support informed decision-making. Re-biopsy should be performed in cases involving critical chromosomal regions. We have now elaborated on this point in the revised Discussion section (page 11, lines 233-246).

(2) Conclusions: I believe that the advice provided in lines 266-269 of the discussion ("While euploid embryos remain the preferred choice, our findings support the consideration of Seg-A embryos in certain clinical scenarios. Refining selection criteria by the help of parental karyotyping or re-biopsy and developing more reliable embryo assessment methods will be critical in optimizing outcomes in assisted reproductive technologies") is one of the most important take-home messages of this paper. In my opinion, this should be stated forcefully in the conclusions.

Response:

Thank you for highlighting the importance of this recommendation. We fully agree that this is a key take-home message of our study. In response to your suggestion, we have revised the Conclusions section to more explicitly and forcefully state the clinical implications of Seg-A embryo management, particularly emphasizing that while euploid embryos remain the preferred and most reliable choice for embryo transfer, our findings support the careful consideration of Seg-A embryos in selected clinical scenarios, especially when no euploid embryos are available. Key strategies, including parental karyotyping, confirmatory re-biopsies, and the development of more accurate embryo assessment tools, are essential in refining embryo selection and improving reproductive outcomes. These results reinforce the need for individualized counseling and thoughtful decision-making in assisted reproductive technologies, rather than routine exclusion of embryos with segmental abnormalities. We have added these important messages to the Conclusion section (page 14, lines 420-431).

Reviewer 2 Report

Comments and Suggestions for Authors

In manuscript ijms-3568232 the authors report a study aimed at verifying the suitability to implantation of IVF-obtained embryos showing segmental aneuploidy. The authors show that this anomaly might still be compatible with a successful, euploid fetus formation, at least to some extent. This result enlarges the number of IVF embryos for implantation in couples wanting a baby. Overall, the result is interesting, but some adjustments are required before publication.

In Table 1, the information provided is incomplete, especially for non-specialists. Please add abbreviation meaning (BMI, AMH, SART) and define (either here or in the text) the following: advanced maternal age (how many years?); repeated miscarriages (how many?); repeat implantation failure (how many?); male factor (what?); others (what?). In addition, “total blastocysts" should be plural; the sentence “Original biopsy on Day 5 biopsy” is unclear or internally redundant, please check.

In figure 2, the a-k statistical test is unclear. As it is now, it seems (for example) that the authors applied 5 times this test to the first table cell (a,b,c,d,e); what is the meaning of two “a” in cells 1 and 2, row 1? And so on. The authors should better clarify how they did the stats and change text and table accordingly.

In figure 3 caption, the last two lines are unclear for both position (they seem like a title) and meaning (do they add some info to the figure?).

In section 4.10 the authors wrote that they used also the ANOVA test. This test is not reported in figures or tables; thus, it is not clear where they used it. In figure 2 (concordance) they report only the chi-test.

The authors should stress in the discussion that using Seg-A embryos should always be a secondary choice, in case there is no way to have euploid embryos for implant.

There are several errors (grammar, typos, unclear sentences) that should be checked by a professional English speaker. A (incomplete) list of lines needing a check follows: 30; 46; 53; 56; 89; 110; 216; 235; 243; 258.

Author Response

Dear reviewer 2,

We sincerely appreciate your valuable comments. Please find below our responses and the corresponding revisions made to the manuscript. In response, we have carefully revised the manuscript and addressed all comments in a detailed point-by-point reply.
Comments of Review 2

In manuscript ijms-3568232 the authors report a study aimed at verifying the suitability to implantation of IVF-obtained embryos showing segmental aneuploidy. The authors show that this anomaly might still be compatible with a successful, euploid fetus formation, at least to some extent. This result enlarges the number of IVF embryos for implantation in couples wanting a baby. Overall, the result is interesting, but some adjustments are required before publication.

In Table 1, the information provided is incomplete, especially for non-specialists. Please add abbreviation meaning (BMI, AMH, SART) and define (either here or in the text) the following: advanced maternal age (how many years?); repeated miscarriages (how many?); repeat implantation failure (how many?); male factor (what?); others (what?). In addition, “total blastocysts" should be plural; the sentence “Original biopsy on Day 5 biopsy” is unclear or internally redundant, please check.

Response: Thank you for your valuable comments regarding Table 1. We have revised the table accordingly and provide detailed responses below:

  1. Abbreviations (BMI, AMH, SART):
    We have added the full definition for SART (Society for Assisted Reproductive Technology) to the text. We have also added the full definitions for BMI (Body Mass Index) and AMH (Anti-Müllerian Hormone) in the footnotes of Table 1 to enhance clarity for non-specialist readers.
  2. Definition of Advanced Maternal Age:
    We have clarified that advanced maternal age refers to women ≥38 years old, and this has been specified directly in the table.
  3. Definition of Repeated Miscarriage:
    This refers to women with ≥2 pregnancy losses, and this definition has been added to the table for clarity.
  4. Definition of Repeat Implantation Failure:
    Defined as ≥3 failed embryo transfers, now explicitly stated in the table.
  5. Definition of Male Factor:
    We have clarified this indication as severe oligospermia, which has been specified in the table.
  6. Clarification of “Others”:
    The category "Others" has been further specified to include cases with unexplained infertility, tubal factors, and cases undergoing PGT for monogenic disorders and aneuploidy. This has been noted in the table footnote.
  7. Correction of “total blastocysts”:
    The term has been corrected to "Total blastocysts" (plural).
  8. Clarification of “Original biopsy on Day 5 biopsy”:
    We agree that this phrasing was unclear. It has been revised to "Biopsied on Day 5" and "Biopsied on Day 6" for clarity and conciseness.

We appreciate the reviewer’s careful attention to detail, which has improved the clarity and completeness of our table (page 3, lines 96-98; page 4, Table 1).

In figure 2, the a-k statistical test is unclear. As it is now, it seems (for example) that the authors applied 5 times this test to the first table cell (a,b,c,d,e); what is the meaning of two “a” in cells 1 and 2, row 1? And so on. The authors should better clarify how they did the stats and change text and table accordingly.

Response:

Thank you for your valuable comment regarding the statistical annotations in Figure 2. We appreciate the opportunity to clarify these annotations. All statistical comparisons in Figure 2 were conducted using the Chi-square test to evaluate differences in categorical outcomes between groups. The superscript letters (a–k) denote statistically significant pairwise differences between specific groups. Each letter represents a unique comparison. When the same letter appears in two different cells, this indicates a significant difference between those two values (P < 0.05). For example, the superscript a in both the “ICM-EU: 85.0%” (TE: EU group) and “ICM-Seg-A: 7.5%” cells indicates a statistically significant difference between those two values. The presence of multiple letters in a single cell (e.g., a, b, c) means that the value significantly differs from multiple other groups, each denoted by a corresponding superscript. We have revised the figure legend to clarify this for readers (page 5, Figure 2).

In figure 3 caption, the last two lines are unclear for both position (they seem like a title) and meaning (do they add some info to the figure?).

Response:

Thank you for your helpful comment. We apologize for the confusion caused by the last two lines of the original figure legend. Our intention was to provide a brief concluding remark summarizing the main message of the figure, namely the concordance and discordance observed between TE and ICM in segmental abnormalities and mosaicism. To improve clarity and avoid misunderstanding, we have revised the legend by rephrasing this summary sentence and repositioning it clearly at the end of the caption. We believe that the updated version better communicates our intent without disrupting the structure or flow of the figure description (page 6, lines 136-138; Figure 3)

In section 4.10 the authors wrote that they used also the ANOVA test. This test is not reported in figures or tables; thus, it is not clear where they used it. In figure 2 (concordance) they report only the chi-test.

Response:

Thank you for pointing this out. We acknowledge the inconsistency in the Methods section. Although we initially considered ANOVA during the statistical planning stage, all actual analyses presented in Figure 2 and related comparisons were performed using the Chi-square test, which is more appropriate for categorical data such as chromosomal classification. We have revised Section 4.10 to remove the mention of ANOVA and to clarify that only Chi-square tests were used for the group comparisons shown in the figures and tables. We appreciate your careful examination of the text, which has helped to improve the clarity and accuracy of our methods description (page 14, lines 413-417).

The authors should stress in the discussion that using Seg-A embryos should always be a secondary choice, in case there is no way to have euploid embryos for implant.

Response:

Thank you for the insightful suggestion. In response, we have revised the Discussion section to more clearly emphasize that Seg-A embryos should be considered only as a secondary option. Although our data and other reports suggest that a subset of Seg-A embryos may have reproductive potential, it is important to emphasize that euploid embryos should always remain the first-line choice for transfer due to their significantly higher predictability and implantation outcomes. The transfer of Seg-A embryos should only be considered as a secondary option, particularly in cases where no euploid embryos are available for implantation. This approach aligns with current PGT-A recommendations, which advocate for cautious use of segmental abnormal embryos given their uncertain developmental competence and lower concordance with ICM results (page 12, lines 317-324).

There are several errors (grammar, typos, unclear sentences) that should be checked by a professional English speaker. A (incomplete) list of lines needing a check follows: 30; 46; 53; 56; 89; 110; 216; 235; 243; 258.

Response:

Thank you for your helpful comments regarding the language and clarity. We have carefully reviewed and revised the manuscript for grammar and typographical errors and sentence clarity, especially focusing on the sections you highlighted (lines 30, 46, 53, 56, 89, 110, 216, 235, 243, and 258). The manuscript has now been edited by a native English speaker with academic writing experience. We believe that the revised version has a significantly improved clarity and language quality. We sincerely appreciate your feedback, which helped improve the readability of our manuscript.

Round 2

Reviewer 1 Report

Comments and Suggestions for Authors

I thank the authors for their kind replies and for heeding my advice. I have no more concerns.

Reviewer 2 Report

Comments and Suggestions for Authors

I believe that the authors satisfactorily replied to my concerns.